# Structural Integrity Assessment of an NEPE Propellant Grain Considering the Tension–Compression Asymmetry in Its Mechanical Property

**DOI:** 10.3390/polym15163339

**Published:** 2023-08-08

**Authors:** Pengjun Zhang, Wangshen Han, Dongmo Zhou, Hanxu Wu

**Affiliations:** School of Mechatronic Engineering, North University of China, Taiyuan 030051, China; zhangpj@nuc.edu.cn (P.Z.); hwswh666@163.com (W.H.); 13832095120@163.com (H.W.)

**Keywords:** structural integrity, NEPE propellant, tension–compression asymmetry, mechanical property, safety factor

## Abstract

In order to investigate the effect of tension–compression asymmetry of propellant mechanical properties on the structural integrity of a Nitrate Ester Plasticized Polyether (NEPE) propellant grain, the unified constitutive equations under tension and compression were established, a new method for grain structural integrity assessment was proposed and the structural integrity of the NEPE propellant grain under the combined axial and transverse overloads was evaluated. The results indicate that the mechanical state of the NEPE propellant grain is in the coexistence of tension and compression under the combined axial and transverse overloads, and the tension and compression regions in the propellant grain is independent of the propellant constitutive behavior. The tension–compression asymmetry of the propellant mechanical properties has a certain impact on its mechanical response. The maximum equivalent stress and strain considering the tension–compression asymmetry falls between that obtained through the tension and compression constitutive model, and is the same as damage coefficient. The safety factor of the NEPE propellant grain considering the tension–compression asymmetry of its mechanical properties is larger than that non-considering, and the traditional method of structural integrity assessment is conservative.

## 1. Introduction

Composite solid propellant (CSP) grain is an important component of a solid rocket motor (SRM), and its structural integrity is crucial for the safe use of the SRM [1]. Currently, the mechanical models of propellant materials used in the structural integrity assessment of propellant grain are mostly based on the mechanical test of a single mechanical state, such as uniaxial tension or compression [2,3,4,5]. CSP grains experience various stresses and strains throughout their life cycle, including manufacturing, transportation, storage, and flight. These conditions lead to complex internal stress states, where the most common is the coexistence of both tensile and compressive stresses [6]. The mechanical properties of CSPs exhibit significant tension–compression asymmetry. Therefore, it is necessary to consider the tension–compression asymmetry of propellant material’s mechanical properties in the structural integrity assessment of propellant grain.

The fundamental reason for the tension–compression asymmetry in the mechanical properties of CSPs lies in their microstructure, which leads to different mechanical response mechanisms under tensile and compressive loads. The adhesive system of a CSP is a high-performance material, whose strength under compressive loading is higher than that under tensile loading due to the reduced mobility of its chain segments. As a particulate filler material, CSPs’ mechanical properties are primarily influenced by the adhesive matrix’s mechanical properties under tensile loading, whereas its mechanical properties are affected by the solid particles’ mechanical properties under compressive loading. This results in lower strength of the propellant material under tensile loads [6].

In recent years, the tension–compression asymmetry in mechanical properties of CSPs has been extensively studied. Majerus and Tamekuni [7] investigated the nonlinear mechanical behavior of a CSP and found it had significantly different properties in tension than in compression. They, therefore, suggested that the nonlinearity due to the strain magnitude and stress state (tension vs. compression) should be taken into account for certain design problems. Ren et al. [8] conducted the tension/compression comparison tests on CSP at different strain rates; the results showed that the compression modulus was higher than the tensile modulus under the same strain rate and the variation in compression modulus with compression velocity was consistent with that of the tensile modulus. Wang et al. [9] demonstrated that the compressive strength of a Hydroxyl-Terminated Polybutadiene (HTPB) propellant was significantly greater than its tensile strength. It is easier for a CSP to fail because of the tensile stress rather than the compressive stress under dynamic loading. Wu et al. [10] investigated the tension and compression mechanical properties of an azide polyether propellant. They observed that the compressive strength of the azide polyether propellant was also greater than its tensile strength. Zhang et al. [11] conducted a comparative experiment of uniaxial tension and compression; they found that temperature and strain rate had a significant effect on the tension–compression modulus ratio of the propellant under low temperature conditions: the higher the strain rate and the lower the temperature, the greater the tension–compression modulus ratio. Wang et al. [12] studied the tension and compression behavior of a modified double-base propellant under a constant strain rate; the results showed that this propellant had asymmetric mechanical effects in tension and compression loads, which were dependent on the strain rate and temperature. In addition, the tension–compression asymmetry properties of other materials have been studied [13,14].

In summary, increasing attention has been paid to the effect tension–compression asymmetry of propellant mechanical properties and many achievements have been obtained in recent years. Unfortunately, these results were rarely applied to the structural integrity assessment of propellant grains.

Therefore, the effect of the tension–compression asymmetry of an NEPE propellant’s mechanical properties on the grain structural integrity was investigated in this paper. The organization of this paper is as follows: Numerical method for tension–compression asymmetry of the NEPE propellant is provided in Section 2. The constitutive model and finite element model are introduced in Section 3. The numerical results and structural integrity of different constitutive models are analyzed in Section 4. Finally, the conclusions are summarized in Section 5.

## 2. Numerical Method

### 2.1. Numerical Method for Tension–Compression Asymmetry

CSPs are typical viscoelastic materials; they have a significant tension–compression asymmetry and are similar to bi-modulus materials. For numerical methods of bi-modulus materials, a typical approach is based on the Ambartsumyan constitutive model, which uses principal stresses to determine the material tension–compression state. Based on this, elastic parameters of the material are selected to construct a stiffness matrix with different modulus for tension and compression along the principal stress direction [15,16]. However, this approach suffers from issues of unstable iteration, low convergence, and poor efficiency. Research suggests that the root cause of these problems lies in the lack of shear modulus in the constitutive model, which results in the inability of the elastic matrix to degenerate to the same modulus in materials with different modulus under tension and compression [17].

Therefore, a novel method for grain structural integrity assessment considering the tension–compression asymmetry of a propellant’s mechanical properties was proposed in this paper. In specific, the average principal stress at every Gaussian point $\sigma_{ai}=(\sigma_{i1}+\sigma_{i2}+\sigma_{i3})/3$ (i=1, 2, ⋯N) is used to determine the tensile/compressive state of the propellant grain and the tensile/compressive constitutive model parameters are assigned accordingly to each Gaussian point of a finite element model (FEM). This method can fully consider the influence of tension–compression asymmetry in a propellant’s mechanical properties on the grain structural integrity, and it has lower element numbers, degrees of freedom and computational workload with good convergence. Furthermore, it preserves the fundamental features of conventional FEMs, such as universality and program consistency, while exhibiting robust flexibility towards intricate geometries and diverse physical quandaries. The specific process is shown in Figure 1.

Define k as the tension and compression discrimination factor. If $\sigma_{ai}\geq 0$, the material is in tension state and k=1; if $\sigma_{ai}<0$, the material is in compression state and k=0. The tension and compression discrimination factor k can be expressed as follows:(1)$$\left\{ \begin{array}{l} \sigma_{ai}\geq 0\;,\;\;\;k=1\\ \sigma_{ai}<0\;,\;\;\;k=0 \end{array}\right.$$

### 2.2. Numerical Methods Verification

A bimodular bar subjected to a tensile force and self-weight, as shown in Figure 2a, is used to verify the proposed method in this paper [18]. In Figure 2a, the rest length of the bar is 10 m, the area of cross section is 1 m × 1 m, the self-weight of the material per unit volume is $\gamma =2.0\;N/m^{3}$, and the load parameters are $P=6.0\;N/m^{2}$. Eighty 8-node brick elements are used to discretize the 3D column. Four cases were analyzed: (i) $E^{+}=E^{-}=5000\;Pa$; (ii) $E^{+}=2500\;Pa$, $E^{-}=5000\;Pa$; (iii) $E^{+}=1000\;Pa$, $E^{-}=5000\;Pa$; and (iv) $E^{+}=500\;Pa$, $E^{-}=5000\;Pa$.

Vertical displacements of nodes with different coordinates z are presented in Table 1, in which $u_{z*}$ denotes the analytical solution of the 1D bimodular problem given via Ambartsumyan. The $u_{z}$ denotes the numerical results of a 3D column that are obtained using the method in this paper. It can be found that the maximum error between the results obtained via the method in this paper and analytical solutions is approximately 1.25%, which confirms the accuracy of the method proposed in this paper.

## 3. Analysis and Modeling

### 3.1. Constitutive Model

Based on the ZWT constitutive model with damage [19,20] and the tension–compression asymmetry judgment criterion in Section 2.1, the unified constitutive equations for the NEPE propellant under tension and compression are presented as follows: (2)$$\begin{array}{c} \sigma =k(1-D)(f_{e}{\left(\varepsilon \right)}^{+}+E_{1}^{+}\int_{0}^{t}\dot{\varepsilon }\;\exp(-\frac{t-\tau }{\theta_{1}^{+}})d\tau +E_{2}^{+}\int_{0}^{t}\dot{\varepsilon }\;\exp(-\frac{t-\tau }{\theta_{2}^{+}})d\tau )+\\ (1-k)(1-D)(f_{e}{\left(\varepsilon \right)}^{-}+E_{1}^{-}\int_{0}^{t}\dot{\varepsilon }\;\exp(-\frac{t-\tau }{\theta_{1}^{-}})d\tau +E_{2}^{-}\int_{0}^{t}\dot{\varepsilon }\;\exp(-\frac{t-\tau }{\theta_{2}^{-}})d\tau ) \end{array}$$
where
(3)$$f_{e}\left(\varepsilon \right)=\sigma_{m}\left[1-\exp\left(-\alpha \varepsilon \right)\right]$$
(4)$$D=\left\{ \begin{array}{l} 0\;\;\;\;\;\;\;\;\;\;\;\varepsilon \leq \varepsilon_{th}\\ 1-\exp[-{\left(\frac{\varepsilon -\varepsilon_{th}}{\eta }\right)}^{m}]\;\;\;\;\;\;\varepsilon \geq \varepsilon_{th} \end{array}\right.$$
where D is the damage variable, $f_{e}(\varepsilon )$ is the nonlinear term of constitutive model, $E_{1}$ and $\theta_{1}$ are the elastic constants and relaxation time of the material at low strain rates, respectively, $E_{2}$ and $\theta_{2}$ are the elastic constants and relaxation time of the material at high strain rates, respectively, $\dot{\varepsilon }$ is the strain rate, $\sigma_{m}$ and α are material parameters, $\sigma_{m}$ represents the limit value of $f_{e}(\varepsilon )$ when the strain approaches infinity, α is the ratio of initial modulus to $\sigma_{m}$ under equilibrium, $\varepsilon_{th}$ is the damage strain threshold, and η and m are scale parameters and shape parameters of Weibull distribution function, respectively.

In Equation (2), when k=1, the propellant material is in a mechanical tensile state, and the nonlinear constitutive model parameters with damage are represented by superscripts with “+”. When k=0, the propellant material is in a mechanical compressive state, and the nonlinear constitutive model parameters with damage are represented by superscripts with “−”.

Refs. [21,22] investigated the tensile and compressive mechanical properties of a NEPE propellant. The test results display that the maximum tensile strength of the propellant is 0.7402 MPa with a maximum elongation of 0.6993, and the maximum compressive strength is 2.078 MPa with a maximum compressive strain of 0.90. The constitutive model parameters under tensile and compressive states were obtained based on the data from Refs. [21,22], as shown in Table 2.

### 3.2. Increment Alization of the Constitutive Equations

Direct implementation of the developed constitutive equations into a finite element program is not feasible. For any constitutive equation used in the finite element formulation, it is necessary to establish the relation between the increment of strain (or deformation rate) and the increment of stress. In order to describe the structural responses of a propellant grain, the incremental constitutive relation was adopted and illustrated as follows.

The form of tension and compression constitutive equation is the same, so only the tension constitutive equation is taken as an example for explanation. The viscoelastic term in the equation is divided into low strain rate viscoelastic term and high strain rate viscoelastic term. The derivation process of high and low strain rate viscoelastic term is consistent, so only the low strain rate viscoelastic term is derived.

The equation is based on decomposing the stress and strain within an isotropic body into deviatoric (shear) and hydrostatic (volumetric) components [23,24,25]:(5)$$\sigma_{ij}=S_{ij}+\frac{1}{3}\delta_{ij}\sigma_{kk}\;\;\;\sigma_{kk}=\sigma_{11}+\sigma_{22}+\sigma_{33}$$
(6)$$\varepsilon_{ij}=e_{ij}+\frac{1}{3}\delta_{ij}\varepsilon_{kk}\;\;\;\varepsilon_{kk}=\varepsilon_{11}+\varepsilon_{22}+\varepsilon_{33}$$
where $\delta_{ij}$ is Kronecker delta (1 for *i* = *j* and 0 for *i* ≠ *j*), $S_{ij}$ and $e_{ij}$ are stress deviatoric tensor and strain deviatoric tensor, respectively. $\sigma_{kk}$ and $\varepsilon_{kk}$ are volumetric stress and volumetric strain, respectively. $\sigma_{ii}$ and $\varepsilon_{ii}$ (*i* = 1, 2, 3) are normal stress and normal strain, respectively, where
(7)$$S_{ij}=2Ge_{ij}$$
(8)$$\sigma_{kk}=3K\varepsilon_{kk}$$
where G=E/2(1+v) and K=E/3(1−2v) are shear modulus and volumetric modulus, respectively. v is Poisson’ s ratio. 

When damage is not considered, the constitutive model without damage can be divided into nonlinear term and viscoelastic integral term in Equation (2). Assuming the propellant material is isotropic, the elastic modulus of the nonlinear term can be obtained using a phenomenological method: (9)$$E(\varepsilon_{v})=\frac{\partial \sigma_{v}}{\partial \varepsilon_{v}}=\sigma_{m}\alpha \;\exp(-\alpha \varepsilon_{v})$$
where $\sigma_{v}$ and $\varepsilon_{v}$ are equivalent stress and equivalent strain, respectively. Substituting Equation (9) into Equations (5), (7) and (8) yields the three-dimensional form of nonlinear term.
(10)$$\sigma_{ij}^{Non}=\frac{E(\varepsilon_{v})}{1+v}e_{ij}^{Non}+\delta_{ij}\frac{E(\varepsilon_{v})}{3(1-2v)}\varepsilon_{kk}^{Non}$$
where $\sigma_{ij}^{Non}$ is the nonlinear part of the three-dimensional stress, $\frac{E(\varepsilon_{v})}{1+v}$ is the shear modulus of nonlinear part, and $\frac{E(\varepsilon_{v})}{3(1-2v)}$ is the volumetric modulus of nonlinear part.

Similarly, the three-dimensional form of the viscoelastic term in Equation (2) is as follows:(11)$$\sigma_{ij}^{Visco}=2G\int_{0}^{t}\frac{\partial e_{ij}^{Visco}}{\partial \tau }\;\exp(-\frac{t-\tau }{\theta })d\tau +\delta_{ij}K\int_{0}^{t}\frac{\partial \varepsilon_{kk}^{Visco}}{\partial \tau }\;\exp(-\frac{t-\tau }{\theta })d\tau$$
where $\sigma_{ij}^{Visco}$ is the viscoelastic part of the three-dimensional stress. $2G\int_{0}^{t}\frac{\partial e_{ij}^{Visco}}{\partial \tau }\;\exp(-\frac{t-\tau }{\theta })d\tau$ is the deviatoric stress of viscoelastic part, and $K\int_{0}^{t}\frac{\partial \varepsilon_{kk}^{Visco}}{\partial \tau }\;\exp(-\frac{t-\tau }{\theta })d\tau$ is the volumetric stress of viscoelastic part.

Assuming that the damage variable is isotropic, the damage variable is a function of the equivalent strain. The damage variable can be expressed as follows:(12)$$D(\varepsilon_{v})=1-\exp[-(\frac{\varepsilon_{v}-\varepsilon_{th}}{\eta }{)}^{m}]$$

Combining Equations (10)–(12), the three-dimensional form of nonlinear viscoelastic constitutive model with damage is as follows:(13)$$\begin{array}{cc} \sigma_{ij,D} & =[1-D(\varepsilon_{v})]\times [\sigma_{ij}^{Non}+\sigma_{ij}^{Visco}]\\ & =[1-D(\varepsilon_{v})]\times [\frac{E(\varepsilon )}{1+v}e_{ij}^{Non}+\delta_{ij}\frac{E(\varepsilon )}{3(1-2v)}\varepsilon_{v}^{Non}\\ & +2G\int_{0}^{t}\frac{\partial e_{ij}^{Visco}}{\partial \tau }\;\exp(-\frac{t-\tau }{\theta })d\tau +\delta_{ij}K\int_{0}^{t}\frac{\partial \varepsilon_{v}^{Visco}}{\partial \tau }\;\exp(-\frac{t-\tau }{\theta })d\tau ] \end{array}$$

Incremental form of the constitutive model is based on the above derivation. Using Equation (10), the incremental form of the nonlinear part can be obtained:(14)$$\Delta \sigma_{ij,Non}^{t+\Delta t}=\sigma_{ij,Non}^{t+\Delta t}-\sigma_{ij,Non}^{t}=\frac{E(\varepsilon_{v})}{1+v}\Delta \varepsilon_{ij,Non}^{t+\Delta t}+\delta_{ij}E(\varepsilon_{v})\lambda \Delta \varepsilon_{v,Non}^{t+\Delta t}$$
where λ=v/[(1+v)(1−2v)].

The increment at time t+Δt is obtained by dividing viscoelastic integral term into shear stress and volumetric stress. The shear stress increment of viscoelastic integral term at time t+Δt is as follows:(15)$$\begin{array}{cc} \Delta S_{ij}^{t+\Delta t} & =S_{ij}^{t+\Delta t}-S_{ij}^{t}\\ & =2G\int_{0}^{t+\Delta t}\frac{\partial e_{ij,Visco}^{t+\Delta t}}{\partial \tau }\;\exp(-\frac{t+\Delta t-\tau }{\theta })d\tau -2G\int_{0}^{t}\frac{\partial e_{ij,Visco}^{t}}{\partial \tau }\;\exp(-\frac{t-\tau }{\theta })d\tau \end{array}$$
where
(16)$$F_{ij}^{t}=\int_{0}^{t}\frac{\partial e_{ij,Visco}^{t}}{\partial \tau }\;\exp(-\frac{t-\tau }{\theta })d\tau$$

Assuming that $\dot{\varepsilon }$ is a constant value at time t+Δt, replaced by the average strain rate, Equation (15) can be expressed as follows:(17)$$\Delta S_{ij}^{t+\Delta t}=2GF_{ij}^{t}[\exp(-\frac{\Delta t}{\theta })-1]+2G\theta \frac{\Delta e_{ij,Visco}^{t+\Delta t}}{\Delta t}[1-\exp(-\frac{\Delta t}{\theta })]$$

Similarly, the volumetric stress increment is as follows:(18)$$\Delta \sigma_{v}^{t+\Delta t}=3KX_{v}^{t}[\exp(-\frac{\Delta t}{\theta })-1]+3K\theta \frac{\Delta \varepsilon_{v,Visco}^{t+\Delta t}}{\Delta t}[1-\exp(-\frac{\Delta t}{\theta })]$$
where $X_{kk}^{t}=\int_{0}^{t}\frac{\partial \varepsilon_{kk,Visco}^{t}}{\partial \tau }\exp(-\frac{t-\tau }{\theta })d\tau$.

Substituting Equation (17) into Equation (18) yields the stress increment form of viscoelastic integral term:(19)$$\begin{array}{cc} \Delta \sigma_{ij,Visco}^{t+\Delta t} & =\Delta S_{ij}^{t+\Delta t}+\frac{1}{3}\delta_{ij}\Delta \sigma_{v}^{t+\Delta t}\\ & =2GF_{ij}^{t}[\exp(-\frac{\Delta t}{\theta })-1]+2G\theta \frac{\Delta e_{ij,Visco}^{t+\Delta t}}{\Delta t}[1-\exp(-\frac{\Delta t}{\theta })]\\ & +\frac{1}{3}\delta_{ij}\{3KX_{v}^{t}[\exp(-\frac{\Delta t}{t})-1]+3K\theta \frac{\Delta \varepsilon_{v,Visco}^{t+\Delta t}}{\Delta t}[1-\exp(-\frac{\Delta t}{\theta })]\} \end{array}$$

Combining Equations (14) and (19), the increment form of nonlinear viscoelastic constitutive model without damage can be expressed as follows: (20)$$\begin{array}{cc} \Delta \sigma_{ij}^{t+\Delta t} & =\Delta \sigma_{ij,Non}^{t+\Delta t}+\Delta \sigma_{ij,Visco}^{t+\Delta t}\\ & =\frac{E(\varepsilon )}{1+v}\Delta \varepsilon_{ij,Non}^{t+\Delta t}+\delta_{ij}E(\varepsilon )\lambda \Delta \varepsilon_{v,Non}^{t+\Delta t}\\ & +2GF_{ij}^{t}[\exp(-\frac{\Delta t}{\theta })-1]+2G\theta \frac{\left(\Delta \varepsilon_{ij,Visco}^{t+\Delta t}-\frac{\Delta \varepsilon_{v,Visco}^{t+\Delta t}}{3}\right)}{\Delta t}[1-\exp(-\frac{\Delta t}{\theta })]\\ & +\delta_{ij}KX_{v}^{t}[\exp(-\frac{\Delta t}{\theta })-1]+K\theta \frac{\Delta \varepsilon_{v,Visco}^{t+\Delta t}}{\Delta t}[1-\exp(-\frac{\Delta t}{\theta })] \end{array}$$

The dynamic damage evolution rate is expressed as the derivative of damage variable with respect to time: (21)$$\frac{dD(\varepsilon )}{dt}={\dot{\varepsilon }}_{v}\frac{m}{\eta }{\left(\frac{\varepsilon_{v}-\varepsilon_{th}}{\eta }\right)}^{m-1}\exp[-(\frac{\varepsilon_{v}-\varepsilon_{th}}{\eta }{)}^{m}]$$

When the time increment tends to be infinity, $dD(\varepsilon_{v})$ can be approximated to $\Delta D(\varepsilon_{v})$, so the incremental form of the damage variable at time t+Δt is as follows: (22)$$\Delta D(\varepsilon_{v})=\Delta {\dot{\varepsilon }}_{v}\frac{m}{\eta }{\left(\frac{\varepsilon_{v}^{t+\Delta t}-\varepsilon_{th}}{\eta }\right)}^{m-1}\exp[-(\frac{\varepsilon_{v}^{t+\Delta t}-\varepsilon_{th}}{\eta }{)}^{m}]$$

Combining Equations (20) and (22) can yield the incremental form of nonlinear viscoelastic constitutive model with damage at time t+Δt: (23)$$\begin{array}{cc} \Delta \sigma_{ij,D}^{t+\Delta t} & =\sigma_{ij,D}^{t+\Delta t}-\sigma_{ij,D}^{t}\\ & =[1-D(\varepsilon_{v}^{t+\Delta t})]\sigma_{ij}^{t+\Delta t}-[1-D(\varepsilon_{v}^{t})]\sigma_{ij}^{t}\\ & =[1-D(\varepsilon_{v}^{t+\Delta t})]\Delta \sigma_{ij}^{t+\Delta t}-\Delta D(\varepsilon_{v}^{t+\Delta t})]\sigma_{ij}^{t} \end{array}$$

The tangent modulus matrix can be expressed as the deviatoric derivative of the stress increment to the strain increment. When considering material damage, the damage variable takes the form of a scalar, so the tangent modulus with damage is as follows:(24)$$C_{ijkl}^{t+\Delta t}=[1-D(\varepsilon_{v}^{t+\Delta t})]\frac{\partial \Delta \sigma_{ij}^{t+\Delta t}}{\partial \Delta \varepsilon_{kl}^{t+\Delta t}}-\sigma_{ij}^{t+\Delta t}\frac{\partial \Delta D(\varepsilon_{v}^{t+\Delta t})}{\partial \Delta \varepsilon_{kl}^{t+\Delta t}}$$
where
(25)$$\Gamma =\frac{\Delta \sigma_{ii}^{t+\Delta t}}{\Delta \varepsilon_{v}^{t+\Delta t}}=\{\frac{E(\varepsilon )}{1+\nu }+E(\varepsilon )\lambda +\frac{\theta (4G+3K)[1-\exp(-\frac{\Delta t}{\theta })]}{3\Delta t}\}$$
(26)$$J=\frac{\Delta \sigma_{ii}^{t+\Delta t}}{\Delta \varepsilon_{jj}^{t+\Delta t}}=\{E\left(\varepsilon \right)\lambda +\frac{\theta (3K-2G)[1-\exp(-\frac{\Delta t}{\theta })]}{3\Delta t}\}$$
(27)$$\Phi =\frac{\Delta \sigma_{ij}^{t+\Delta t}}{\Delta \varepsilon_{v}^{t+\Delta t}}=\{\frac{E(\varepsilon )}{1+v}+\frac{2G\theta [1-\exp(-\frac{\Delta t}{\theta })]}{\Delta t}\}$$

The tangent modulus matrix of the nonlinear viscoelastic constitutive model with damage can be obtained by expanding Equation (24):(28)$$C_{ijkl}^{t+\Delta t}=[1-D^{t+\Delta t}(\varepsilon )]\left[\begin{array}{cccccc} \Gamma & J & J & 0 & 0 & 0\\ J & \Gamma & J & 0 & 0 & 0\\ J & J & \Gamma & 0 & 0 & 0\\ 0 & 0 & 0 & \Phi & 0 & 0\\ 0 & 0 & 0 & 0 & \Phi & 0\\ 0 & 0 & 0 & 0 & 0 & \Phi \end{array}\right]$$

### 3.3. Model Verification

The unified constitutive equations for the NEPE propellant under tension and compression is validated against the experimental data (Figure 3) obtained from Refs. [21,22]. Based on the three-dimensional damage constitutive model derived in Section 3.2, the ABAQUS UMAT (ABAQUS ver. 2022, Simulia; Dassault Systemes, Vélizy-Villacoublay, France) subroutines were developed and the stress–strain curves of the tensile specimens under different confining pressures were predicted. The stress–strain simulation curves of the samples were obtained by selecting point A of the tension sample and point B of the compression sample, and compared with the experimental curves and fitted curves, as shown in Figure 3. It is clear that the prediction results are in good agreement with experimental data and support that the unified constitutive equations reasonably well captures the stress–strain response of the NEPE propellant under tension and compression loads.

### 3.4. Finite Element Modeling

The real model is a tube grain obtainable via case-bonded casting. Due to the symmetry in its geometry, a model of 180° segment with symmetric boundary conditions on the cut face was utilized for simplicity without loss of accuracy. The NEPE propellant, insulation and case were modeled with 105,280 eight-node continuum elements (C3D8R) and 130,464 grid nodes, as shown in Figure 4.

The parameters for structural integrity analysis in these expressions are given in Table 3.

The model is created with the following assumptions and boundary conditions: (i) The thickness of the case was constant. (ii) The insulation liner was elastic. (iii) The outer surface of the case was fixed and subjected to natural convection boundary conditions and the symmetry plane was set with symmetry constraints. (iv) The interfaces between the case/insulation/propellant were set as “Tie” in the ABAQUS software.

The calculation conditions are as follows: The SRM is subjected to the combined action of axial and transverse overload, with both axial and radial overload of 100 g, as shown in Figure 5.

## 4. Results and Discussion

### 4.1. Mechanical Response

When considering the tension–compression asymmetry of the propellant mechanical properties, the distribution of k in the propellant grain under the combined axial and transverse overloads is depicted in Figure 6a, where the red region indicates tension state and the blue region indicates compression state. It is evident that the compressed region is mainly located on the side where the transverse load is directed in propellant grain, while the remaining areas are in a tension state. This means that the propellant grain exhibits two different mechanical states when subjected to the combined axial and transverse overloads.

When examining the mechanical states of the propellant only in terms of tension or compression constitutive model, the boundary between the tension and compression regions of the propellant grain is illustrated in Figure 6b,c, respectively. Interestingly, the boundaries of tension and compression regions in the propellant grain are almost identical under all three mechanical states. This indicates that the boundary between tension and compression regions is independent of the propellant constitutive behavior. 

Figure 7 and Figure 8 present the equivalent stress $\sigma_{V}$ and equivalent strain $\varepsilon_{V}$ of propellant grain under the combined axial and transverse overloads, respectively. It is clear that the critical location is at the root of stress-release boot. The maximum equivalent stress $\sigma_{Vm}$ of the propellant grain obtained via tension–compression asymmetry, and tension and compression constitutive model are 0.386 MPa, 0.408 MPa, and 0.421 MPa, respectively, while the maximum equivalent strain $\varepsilon_{Vm}$ is 0.154, 0.193 and 0.1, respectively.

For further study the effect of tension–compression asymmetric mechanical behavior of the propellant on the mechanical response of the propellant grain under the combined axial and transverse overloads, six paths, as shown in Figure 9, were selected for analysis.

Figure 10 presents the equivalent stress $\sigma_{V}$ of propellant grain along paths 1~6. The shaded area represents the propellant in a tension state when considering tension–compression asymmetry. It can be seen that the $\sigma_{V}$ near the case is generally higher than that near the inner hole. There is no significant difference between the $\sigma_{V}^{T}$ (the equivalent stress obtained via the tension constitutive model) and $\sigma_{V}^{C}$ (the equivalent stress obtained via the compression constitutive model), except the critical location. However, there is a certain difference in the $\sigma_{V}^{T-C}$ (the equivalent stress obtained via the tension–compression asymmetry unified constitutive model) when considering the tension–compression asymmetry.

Figure 11 presents the equivalent strain $\varepsilon_{V}$ of the propellant grain along paths 1~6. It is clear that the $\varepsilon_{V}$ near the case is also higher than that near the inner hole. The $\varepsilon_{V}^{T}$ (the equivalent strain obtained via the tension constitutive model) is generally greater than $\varepsilon_{V}^{C}$ (the equivalent strain obtained through the compression constitutive model) as a whole due to the tensile modulus of the propellant being less than the compressive modulus; the $\varepsilon_{V}^{T-C}$ (the equivalent strain obtained via the tension–compression asymmetry unified constitutive model) falls between $\varepsilon_{V}^{T}$ and $\varepsilon_{V}^{C}$, at the interface between the tension and compression regions.

The strain response of the propellant grain is determined by the load and the modulus of the propellant, while the stress depends upon the load, the structural, and the boundary conditions and so on [26]. Therefore, the difference in stress response of the grain is relatively small when using the tension or compression constitutive model, while the difference in strain response is relatively large, as shown in Figure 10 and Figure 11, respectively. When considering the tension–compression asymmetry of the propellant mechanical property, the stress of the grain is redistributed due to the large difference in tension and compression moduli, resulting in differences in the stress–strain responses, which also vary with changes in the tension and compression regions.

### 4.2. Structural Integrity Analysis

Damage coefficient was defined to characterize the damage of the propellant grain; damage coefficient $\omega =\varepsilon_{V}/\varepsilon_{th}$. The propellant appears to be damaged when ω≥1, but no damage occurs when ω<1. Figure 12 depicts the variation curve of the damage coefficient ω along paths 1~6. It is obvious that there is no obvious damage in the propellant under the combined axial and transverse overloads. However, the $\omega_{T}$ (the damage coefficient obtained based on the tension constitutive model) is the smallest and the $\omega_{C}$ (the damage coefficient obtained based on the compression constitutive model) is maximal. The $\omega_{T-C}$ (the damage coefficient obtained via the tension–compression asymmetry unified constitutive model) falls between $\omega_{T}$ and $\omega_{C}$, except in path 4.

The strength failure is the main failure mode of the NEPE propellant. To assess the structural integrity of the propellant grain, the safety factor S was introduced and defined as $S=\sigma_{m}/\sigma_{V}$. Von Mises stress criterion was used to analyze the grain structural integrity. Figure 13 presents the safety factor contours of the propellant grain. The results indicate that the critical points of the propellant grain based on the safety factors of three constitutive model are different, that is, the critical points of the propellant grain is dependent on its constitutive model.

The safety factors of the propellant grain under the combined axial and transverse overloads obtained via different constitutive models are shown in Table 4. It can be seen that the safety factor obtained via the tension constitutive model $S_{T}$ is 1.73 and its safety margin is low, while the safety factor obtained via the compression model $S_{C}$ is 4.98; this is due to the fact that the compressive strength is much higher than the tensile strength. However, the safety factor based on the tension–compression asymmetry unified constitutive model $S_{T-C}$ is 5.68. It can be concluded that the mechanical constitutive state of the propellant has a significant effect on the grain structural integrity, and the traditional method of structural integrity assessment based on the tension constitutive model is conservative.

## 5. Conclusions

A numerical calculation method considering the tension–compression asymmetry of the propellant mechanical properties was established and the structural integrity of an NEPE propellant grain under the combined axial and transverse overloads was investigated. The results are as follows:
(1)The mechanical state of the NEPE propellant grain is the coexistence of tension and compression under the combined axial and transverse overloads, and the tension and compression regions in the propellant grain is independent of the propellant constitutive behavior.(2)The tension–compression asymmetry of the propellant mechanical properties has a certain impact on its mechanical response. The maximum equivalent stress and strain considering the tension–compression asymmetry falls between that obtained with the tension and compression constitutive model, and is the same as damage coefficient.(3)The safety factor of the NEPE propellant grain considering the tension–compression asymmetry of its mechanical properties is larger than that non-considering, and the traditional method of structural integrity assessment is conservative.

## Figures and Tables

**Figure 1 polymers-15-03339-f001:**
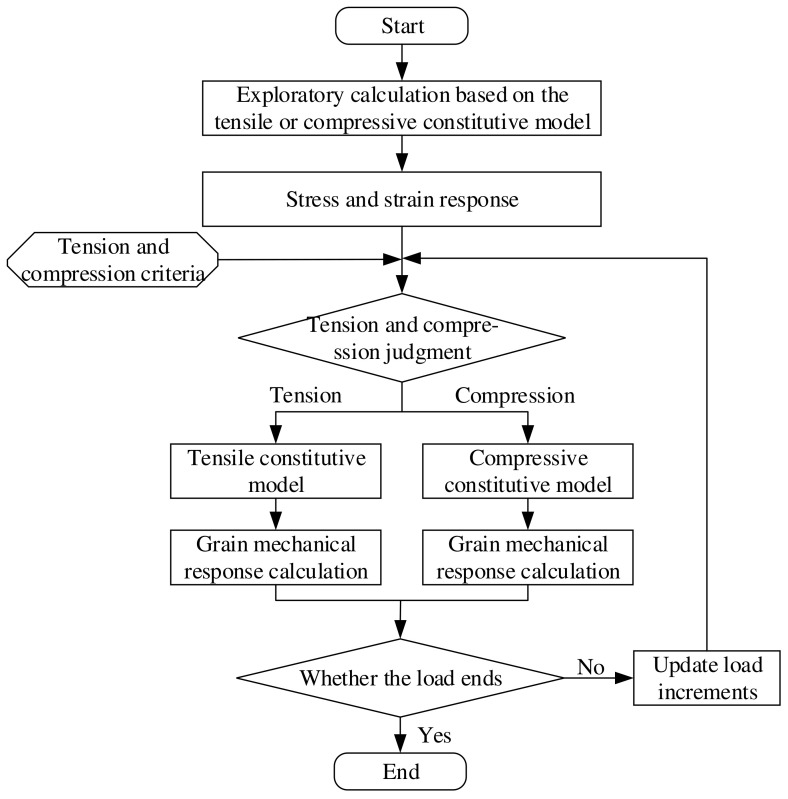
Evaluation process of grain structural integrity.

**Figure 2 polymers-15-03339-f002:**
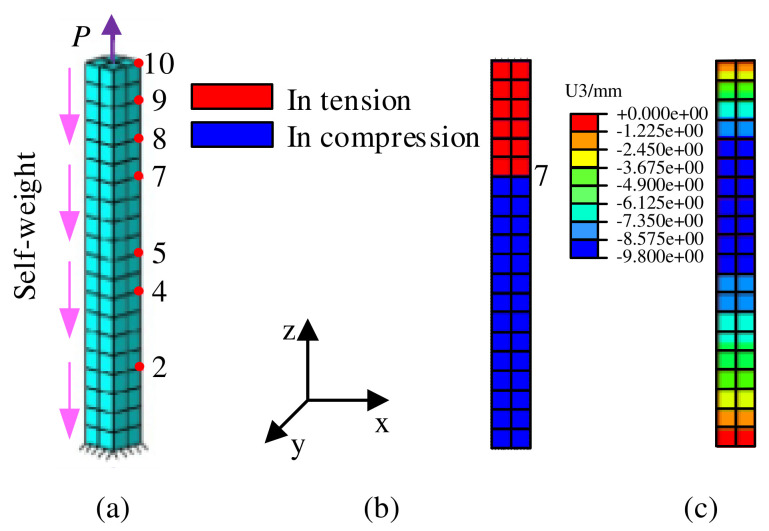
Numerical methods verification model: (**a**) 3D FEM, (**b**) distribution of tension and compression states in case (i), and (**c**) vertical displacement contour of case (i).

**Figure 3 polymers-15-03339-f003:**
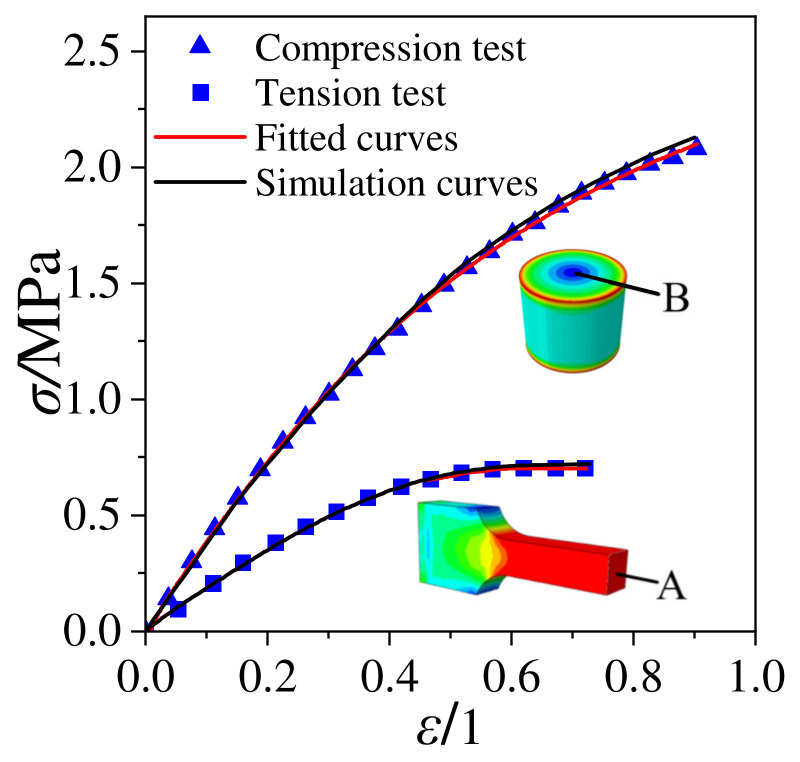
Comparison of experimental curves with numerical prediction results.

**Figure 4 polymers-15-03339-f004:**
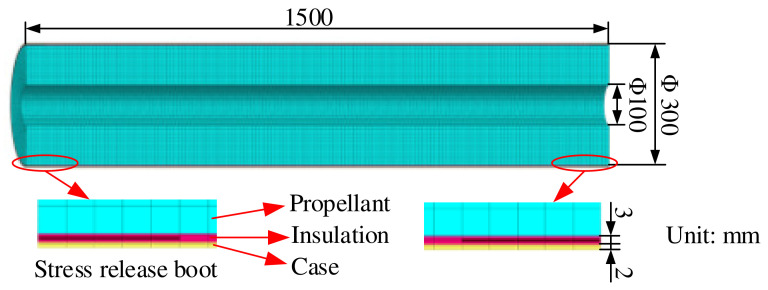
Motor model and mesh.

**Figure 5 polymers-15-03339-f005:**
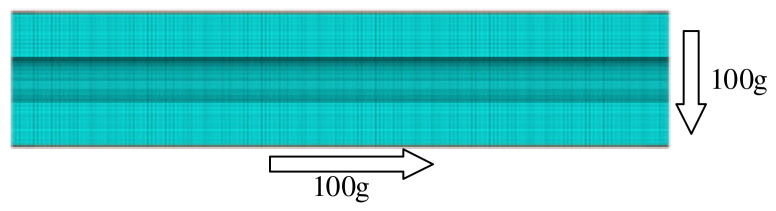
Schematic diagram of load application.

**Figure 6 polymers-15-03339-f006:**
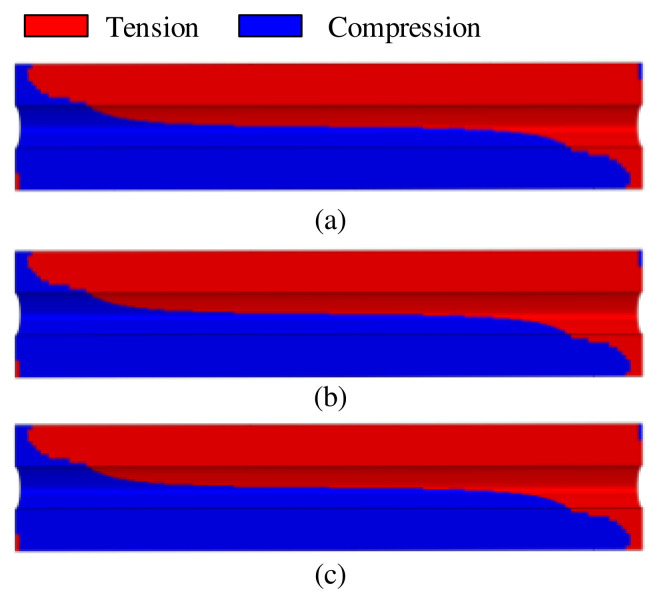
Distribution of tension and compression states: (**a**) tension–compression asymmetry unified constitutive model, (**b**) tension constitutive model, and (**c**) compression constitutive model.

**Figure 7 polymers-15-03339-f007:**
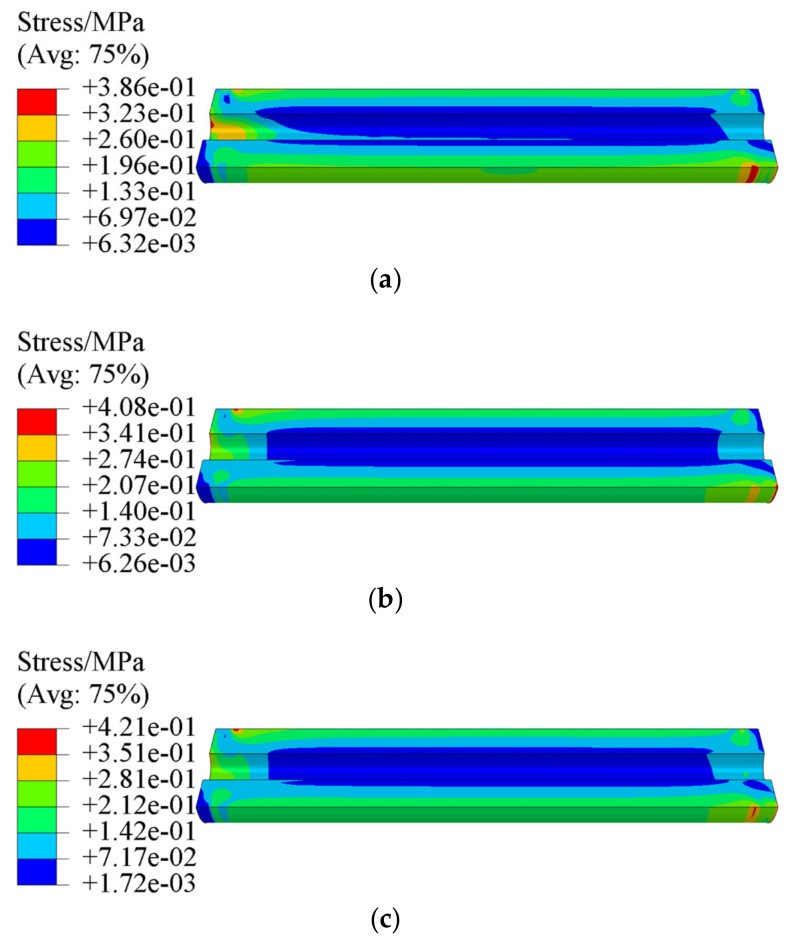
Equivalent stress contours of the grain: (**a**) tension–compression asymmetry unified constitutive model, (**b**) tension constitutive model, and (**c**) compression constitutive model.

**Figure 8 polymers-15-03339-f008:**
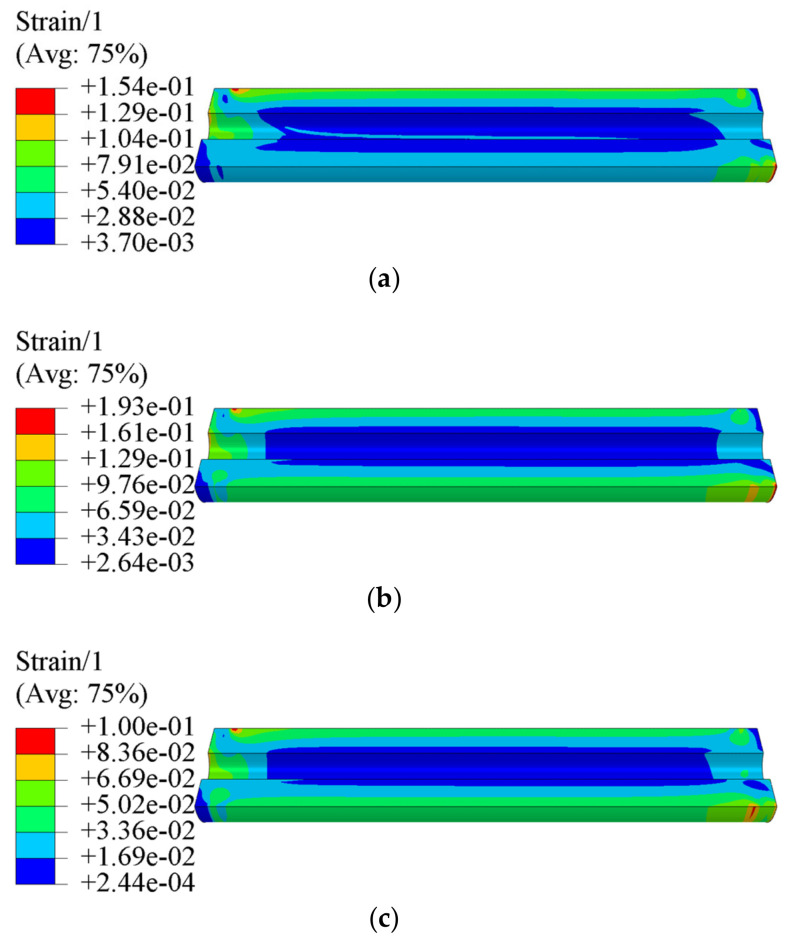
Equivalent strain contours of the grain: (**a**) tension–compression asymmetry unified constitutive model, (**b**) tension constitutive model, (**c**) compression constitutive model.

**Figure 9 polymers-15-03339-f009:**
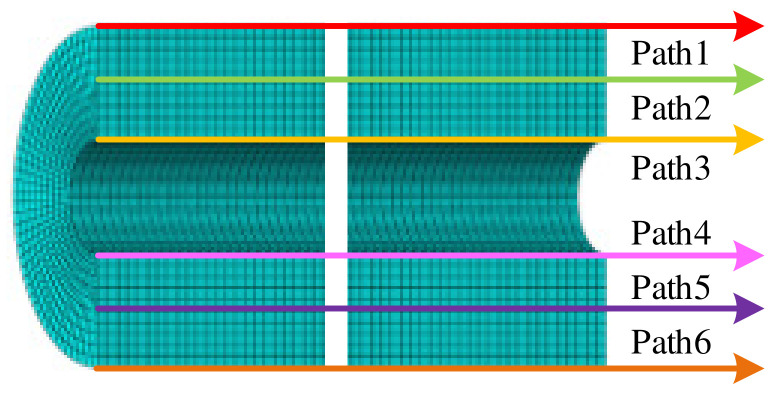
Path selection.

**Figure 10 polymers-15-03339-f010:**
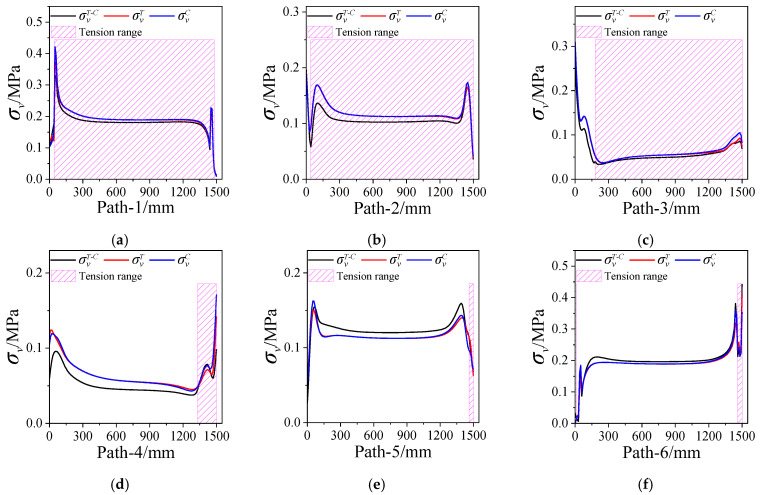
Equivalent stresses vary along paths 1~6. (**a**) Path-1. (**b**) Path-2. (**c**) Path-3. (**d**) Path-4. (**e**) Path-5. (**f**) Path-6.

**Figure 11 polymers-15-03339-f011:**
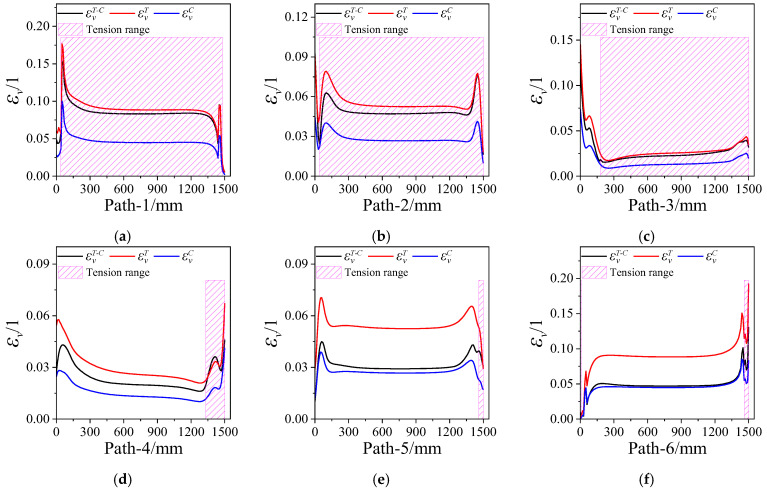
Equivalent strains vary along paths 1~6. (**a**) Path-1. (**b**) Path-2. (**c**) Path-3. (**d**) Path-4. (**e**) Path-5. (**f**) Path-6.

**Figure 12 polymers-15-03339-f012:**
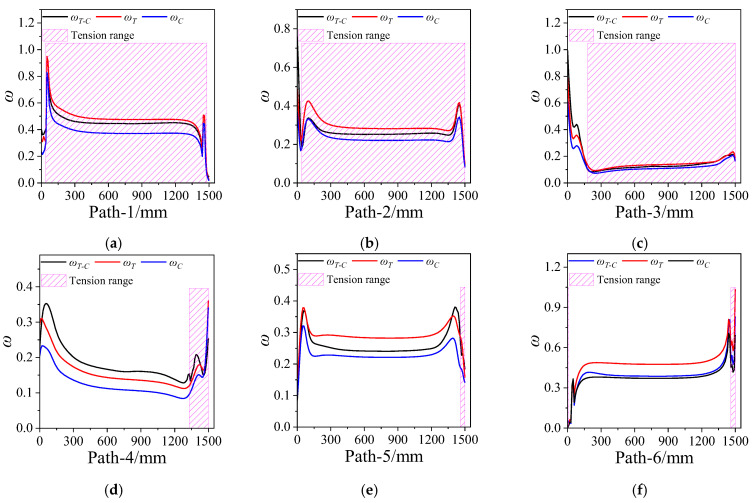
Damage coefficient varies along paths 1~6. (**a**) Path-1. (**b**) Path-2. (**c**) Path-3. (**d**) Path-4. (**e**) Path-5. (**f**) Path-6.

**Figure 13 polymers-15-03339-f013:**
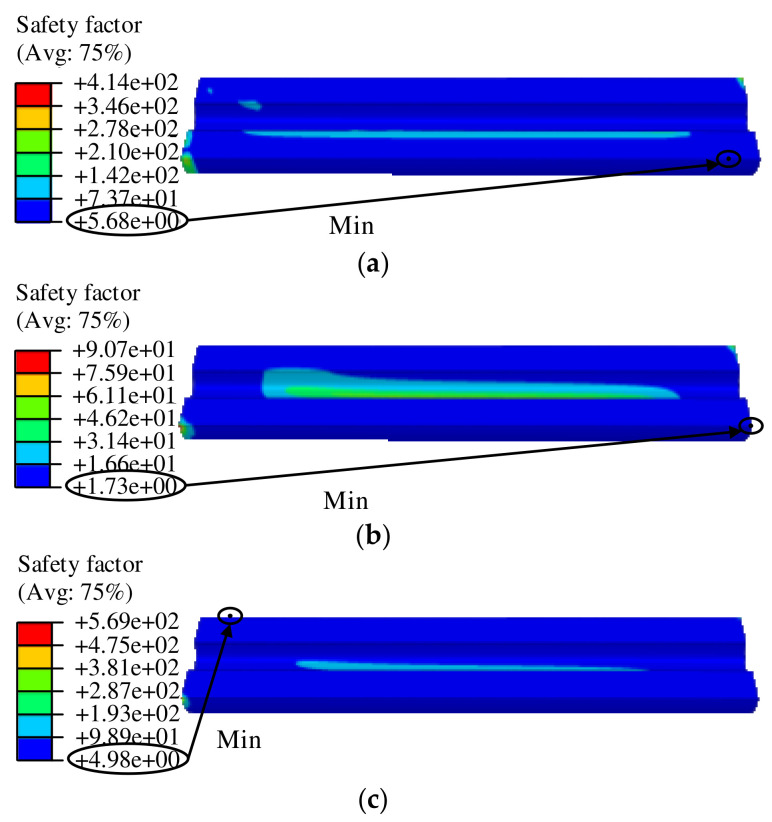
Safety factor based on the equivalent stress criterion: (**a**) tension–compression asymmetry unified constitutive model, (**b**) tension constitutive model, and (**c**) compression constitutive model.

**Table 1 polymers-15-03339-t001:** Comparison of analytical and numerical solutions of vertical displacements.

	*E^−^/E^+^ =* 1			*E^−^/E^+^ =* 2			*E^−^/E^+^ =* 5			*E^−^/E^+^ =* 10		
*z* (m)	*u_z*_* (10^−3^ m)	*u_z_* (10^−3^ m)	error (%)	*u_z*_* (10^−3^ m)	*u_z_* (10^−3^ m)	error (%)	*u_z*_* (10^−3^ m)	*u_z_* (10^−3^ m)	error (%)	*u_z*_* (10^−3^ m)	*u_z_* (10^−3^ m)	error (%)
2	−4.80	−4.80	0	−4.80	−4.80	0	−4.80	−4.80	0	−4.80	−4.80	0
4	−8.00	−8.00	0	−8.00	−8.00	0	−8.00	−8.00	0	−8.00	−8.00	0
5	−9.00	−9.00	0	−9.00	−9.00	0	−9.00	−9.00	0	−9.00	−9.00	0
7	−9.80	−9.80	0	−9.80	−9.80	0	−9.80	−9.80	0	−9.80	−9.80	0
8	−9.60	−9.60	0	−9.40	−9.41	0.11	−8.80	−8.81	0.14	−7.80	−7.81	0.13
9	−9.00	−9.00	0	−8.20	−8.21	0.12	−5.80	−5.81	0.17	−1.80	−1.82	1.11
10	−8.00	−8.00	0	−6.20	−6.21	0.16	−0.80	−0.81	1.25	8.20	8.17	0.37

**Table 2 polymers-15-03339-t002:** Constitutive model parameters under different mechanical states.

	*σ_m_*/MPa	*α*/MPa	*E*_1_/MPa	*θ*_1_/s	*E*_2_/MPa	*θ*_2_/s	*m*/1	*η*/1	*εth*/1
Tensile state	6.771	0.2473	0.1334	0.1161	0.4527	1.259	1.798	1.502	0.186
Compressive state	20.002	0.1	1.909	19.476	0.15	228.437	0.989	0.512	0.101

**Table 3 polymers-15-03339-t003:** Material properties parameters.

Parameter	Case	Insulation	Propellant
Density/(kg/m^3^)	7850	1220	1735
Modulus/MPa	1.96 × 10^5^	30	/
Poisson’s ratio/1	0.28	0.498	0.498

**Table 4 polymers-15-03339-t004:** Comparison of safety factor.

Safety Factor	Value
*S_T_*	1.73
*S_C_*	4.98
*S_T−C_*	5.68

## Data Availability

The data that support the findings of this study are available on request from the corresponding author.

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
