# Peer review of "Structural Integrity Assessment of an NEPE Propellant Grain Considering the Tension–Compression Asymmetry in Its Mechanical Property"

_polymers, 2023, doi:10.3390/polym15163339_

Round 1

Reviewer 1 Report

This is a original article on the effect of tension-compression asymmetry of propellant mechanical properties on the structural integrity of Nitrate Ester Plasticized Polyether (NEPE) propellant grain. The authors clearly determined the aim of the study. The introduction presents the scientific problem in a comprehensive manner. However, before allowing for publication, I would suggest making some minor improvements:

- citations must be enclosed in square brackets,

- lines 66 - 67 - there is repetition of "modified double-base propellant" in the sentence, you need to change the style,

- line 107 - "self-weight" - you need to mark the direction of gravity in Figure 2,

- lines 111 - 112 - there were three cases, are figures 2b and 2c for case 1 and 2 ?, and what are the displacement maps for case "c" ?

- figure 2b and 2c shows displacement maps, what is the unit ?

- table 2 - no units for the listed parameters,

- lines 242 - 243 - boundary conditions o loads should be marked in Figure 4 or in a separate drawing,

- line 268, Figures 6 and 9 - large "P" at [MPa],

- is the caption under figure 4 the correct "Motor model" ?

- there are no dimensions for the analyzed model as shown in Figure 4,

- the attractiveness of the work could be enhanced by photos of the real propellant grain

After completing the above mentioned comments, I recommend the paper for publication.

Reviewer 2 Report

Zhang et al. investigated the effect of tension-compression asymmetry of propellant mechanical properties on the structural integrity of NEPE propellant grain. For this purpose, they developed unified constitutive equations under tension and compression. The proposed computational method was validated using published experimental results. The paper can be accepted for publication after addressing the following comments:

1) Autors should carefully check the references. For example, this reviewer did not find references 3, 4, 5, 20, and 21 within the text. Also, please put all the references in brackets.

2) Please define all the abreviation including FEM, HTPB

3) Please provide reference for  ZWT constitutive model (line 126)

4) Please add load direction to figure 5 for better reader expirience

5) Figures 6 and 7. Please bring colormaps for parts a, b, and c to a unified form

6) For strains (figure 10), the values computed by tension-compression constitutive model are close to those obtained by tension constitutive model in tension region (Path 1) and by compression constitutive model in compression region (Path 6) which looks logical. However, equivalent stress computed by tension-compression constitutive model in any case lower than those computed by tension or compression constitutive models. More over, the last two a very close to each other. This result is puzzling, and if it not a mistake it deserves more discussion than was made in the current version of the paper 

line 50: It seems that there should be "They"  instead of "He"
